# PolygoNet: Leveraging Polygonal Contours for Efficient Image Classification with deep neural networks

## Abstract

In recent years, deep learning models have demonstrated remarkable capabilities in various image-related tasks, yet they are often plagued by computational complexity and susceptibility to overfitting. In this paper, we propose a novel approach that leverages efficient polygon representation through dominant points for the input images to address these challenges for image classification tasks. Our method focuses on transforming input images into polygon representations, which are subsequently utilized for training deep neural networks. The key contribution lies in the use of theses dominant points, which offer a concise and flexible representation of images. By transforming images into dominant points, we significantly reduce the computational burden associated with processing large image datasets. This reduction in calculation not only accelerates the training process but also conserves computational resources, making our approach particularly appealing for real-time applications and resource-constrained environments. We validate our approach through extensive experiments on benchmark datasets, showcasing its effectiveness in reducing computation. The experimental results demonstrate that our method achieves state-of-the-art performance across various image classification tasks, underscoring its potential on standard configuration and edge computing configuration. The code for the experiments of the paper are provided at https://anonymous.4open.science/r/PolygoNet-7374.

## 1 Introduction

In the burgeoning field of image classification, the efficiency of data representation and processing plays a crucial role, especially in scenarios demanding real-time analysis and decision-making on limited-resource platforms. Traditional image classification methods, which rely on raw image data, often face challenges due to high computational costs and substantial memory requirements, particularly with high-resolution images. The challenge intensifies when dealing with high-resolution images where traditional pixel-based methods become computationally expensive and less feasible for real-time applications. This highlights an essential quest in contemporary AI research: developing methods that not only simplify data complexity but also retain the essential features necessary for accurate analysis. To address these limitations, we propose an novel approach that utilizes dominant points extracted from image contours as a compact, yet effective representation for classification tasks.

Our methodology diverges from conventional practices by implementing an implicit form of image classification. Instead of directly analyzing pixel-level data, our approach focuses on geometrically salient features of objects captured through their contours. This shift from explicit to implicit data representation is achieved through the Modified Adaptive Tangential Cover (MATC) Ngo et al. (2017); Ngo (2019), which identifies dominant points that succinctly encapsulate the essential shape information of objects within the image.

The extraction of dominant points offers a streamlined yet powerful representation, reducing the data dimensionality dramatically while preserving the critical geometric attributes necessary for effective classification. By focusing on these points, our model can efficiently process high volumes of data with reduced computational overhead, making it well-suited for applications on devices with limited processing capabilities. Furthermore, this approach enhances the model's ability to generalize from minimal data by focusing on

the structural essence of the images rather than getting potentially misled by background noise or color variations that do not contribute to object recognition.

Moreover, the use of dominant points aligns with the cognitive processes observed in human visual perception, where the human eye tends to recognize shapes and objects based on key structural features rather than the full pixel-by-pixel analysis Biederman (1987); Koffka (2013). This biomimetic aspect of our methodology not only improves the efficiency of the classification process but also potentially increases its accuracy by mimicking more closely how humans perceive and categorize visual information.

In summary, our proposed method stands out by providing a robust solution to the challenges posed by high-resolution image classification. By applying the principles of MATC for dominant point extraction and leveraging a more abstract but representative data form, our approach promises substantial improvements in speed and efficiency for real-time image classification tasks, paving the way for new advancements in edge computing and mobile AI applications.

## 2 Related work

**Image Classification.** is a fundamental task in computer vision where the objective is to assign a predefined label to an image. Deep learning architectures for image classification tasks are primarily based on convolutional neural networks (ConvNets). Since the breakthrough of AlexNet (Krizhevsky et al., 2012), ConvNets have emerged as the predominant architectural choice for computer vision (Simonyan & Zisserman, 2014; Szegedy et al., 2015; He et al., 2016b; Tan & Le, 2019). Meanwhile, with the success of self-attention models such as Transformers Vaswani et al. (2017) in Natural language processing (Brown et al., 2020; Devlin et al., 2018), there has been a notable trend in prior research endeavours to integrate the efficacy of attention mechanisms into Computer Vision models (Wang et al., 2018; Bello et al., 2019; Srinivas et al., 2021; Shen et al., 2021). Noteworthy among these efforts is the recent work on Vision Transformer (ViT) (Dosovitskiy et al., 2020), which illustrates the compelling results achievable through the use of vanilla Transformer layers.

**Shape and contour analysis.** Early efforts in contour classification relied on handcrafted features to represent shapes. Approaches such as Shape context (Belongie et al., 2002) and Fourier descriptors (Kuhl & Giardina, 1982) are classical methods that capture global and local information in contours. These approaches laid the foundation for contour representation and classification, focusing on the extraction of discriminative features from the contour only. With the rise of deep learning, several studies have explored the application of neural networks to contour and shape classification, CNNs have been adapted to process contour information Baker et al. (2018; 2020), demonstrating improved performance in tasks such as handwritten digit recognition and object classification based on boundary information. These approaches leverage the hierarchical features learned by deep networks for effective contour representation.

**Self-attention mechanism.** It represents the core building block of the Transformers architecture which allows the model to learn attention patterns over its input tokens without being limited in extent to a local receptive field as with CNNs. The first use of Attention module was proposed by (Bahdanau et al., 2014) for the neural machine translation, it allows the model to assess the importance of individual tokens within the input sequence relative to the others and to integrate information that can be far away from the current token. This capability enhances the model's capacity to grasp long-range contextual information more effectively. Since then, attention mechanisms have been successfully applied in various natural language processing tasks, including image captioning (Xu et al., 2015) and sentiment analysis. While attention mechanisms initially gained prominence in natural language processing, their application has extended to computer vision tasks. (Wang et al., 2018) introduced a new approach that combines CNN-like architecture while leveraging self-attention for capturing long-range dependencies in images. This is achieved through non-local operations that compute the response at a position as a weighted sum of the features at all positions. This enables the network to account for global information, which is beneficial for several computer vision tasks considered by the authors. The work of (Dosovitskiy et al., 2020) adapted the Transformer architecture to vision, where the image are treated analogous to text sequences. In Vision Transformers (ViT), an image is split into patches, and these patches are processed as if they were words in a sentence. This approach leverages the inherent capability of the Transformer to model interactions between distant elements, thus capturing complex dependencies across the entire image. By using self-attention, ViT can focus on relevant parts of the

image irrespective of their spatial position, enhancing the model's ability to perform a variety of computer vision tasks with remarkable efficiency and accuracy.

**Combination of CNN with self-attention.** The fusion of convolutional neural networks (CNNs) with self-attention mechanisms has sparked significant interest due to its transformative impact across various domains. This innovative approach enhanced image classification by integrating self-attention with CNN feature maps, as outlined by (Bello et al., 2019). Additionally, it has been effectively applied in object detection, as demonstrated in studies by (Hu et al., 2018) and (Carion et al., 2020), and extends to video processing, where (Wang et al., 2018) and (Sun et al., 2019) have made notable contributions. The synergy between CNNs and self-attention also advances image classification techniques, exemplified by (Wu et al., 2020), and facilitates unsupervised object discovery, as seen in the work of (Locatello et al., 2020). Furthermore, this combination has proven crucial in bridging text and vision task, with significant advancements reported by (Chen et al., 2020; Lu et al., 2019; Li et al., 2019), underscoring the broad and impactful applications of merging CNNs with self-attention mechanisms.

Our work leverages this potent combination of self-attention mechanisms with convolutional neural networks. As stated above, self-attention is efficient for combining features that are far apart from the input representation but also presents the interest of being able to deal with variable input size. As detailed in the method section, encoding shapes with dominant points produce variable sized inputs, a complex shape requiring more points than a simple shape to be efficiently encoded.

## 3  Method

### 3.1  Data Preprocessing using Adaptive Tangential Cover

The Modified Adaptive Tangential Cover (MATC) approach is integral to our data preprocessing, particularly for accurately approximating contours (Ngo, 2019). This method is grounded in the concept of blurred segments and tangential cover, defined as a sequence of blurred segments of varying thickness $\nu$, which adjusts dynamically based on local noise levels along a digital curve (Kerautret et al., 2012). The strength of MATC lies in its ability to handle noise and imperfections inherent in digital curves, thus maintaining the integrity and reliability of the approximated contours.

Dominant points, pivotal for representing the geometric characteristics of contours, are identified within the smallest common zones induced by successive blurred segments. These points are characterized by the smallest angle of curvature, making their detection a straightforward process of measuring angles. Our application of MATC begins with the computation of an adaptive tangential cover using a local noise estimator to determine meaningful segment thickness, which dynamically adjusts based on local noise estimations along the curves. This refined approach to determining segment widths, by emphasizing the most frequent meaningful thickness, enhances the handling of localized noise variations and improves accuracy around complex contour details. MATC targets enhancements in the polygonal approximation of digital curves, especially in noisy corners, ensuring a higher fidelity representation. By identifying dominant points within these optimally adjusted segments, we can reconstruct a polygonal representation that captures the essence of the curve's geometry with remarkable fidelity, particularly in areas of intricate detail and significant curvature. The Figure 1 illustrates the preprocessing steps.

Given an input image $I \in \mathbb{R}^{H \times W \times C}$, where H, W, and C represent the height, width, and number of color channels respectively, the process to extract a set of $N$ dominant points, $D$, starts by converting the RGB image to grayscale. This simplifies the data while preserving essential visual information. A threshold is applied to the grayscale image to create a binary image, and filters are used to remove noise and enhance the clarity of shapes. Contours $C = \{c_i \in \mathbb{R}^2\}$ are then extracted from the processed image. The Modified Adaptive Tangential Cover (MATC) process is applied to these contours to identify and extract the different dominant points $D$. It is important to note that the number and positions of these dominant points can vary significantly from one image to another, reflecting the unique features and structural variations present in each image. These dominant points $D$ are represented as a $N \times 2$ matrix where each row corresponds to the $(x, y)$ coordinates of a dominant point on the image plane:

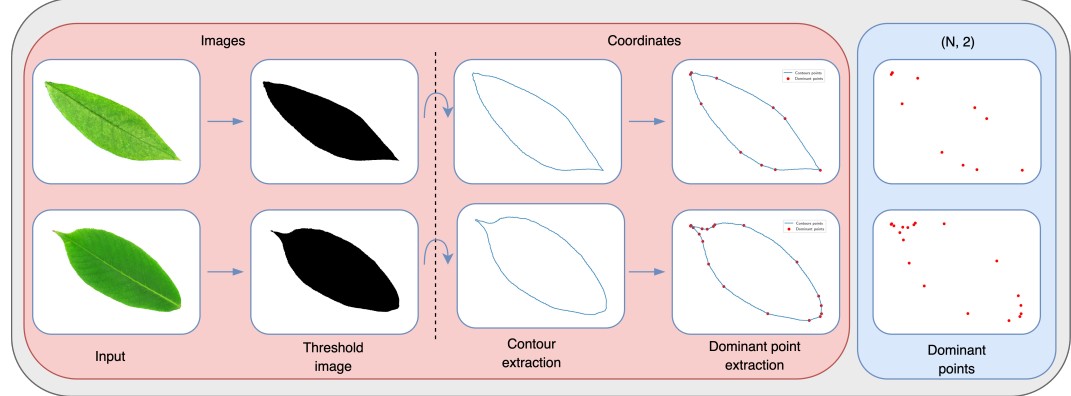

Figure 1: The encoding of the shape is performed by first thresholding the image, extracting its contour, and then computing the dominant points with the Modified Adaptive Tangential Cover algorithm. Note the number of dominants is shape dependent and not the same for every image.

$$D = \begin{bmatrix} x_1 & y_1 \\ x_2 & y_2 \\ \vdots & \vdots \\ x_N & y_N \end{bmatrix}, \quad D \subseteq \mathcal{C}$$

The pseudo-code 3.1 describes the different steps of the data preparation.

---
**Algorithm 1** Extraction of Dominant Points from Image
---
**Require:** Input image $I \in \mathbb{R}^{H \times W \times C}$                    ▷ e.g. Flavia Image size: $(1600 \times 1200 \times 3)$
**Ensure:** Matrix of dominant points $D$ with dimensions $N \times 2$            ▷ Avg dimension of D: $(60 \times 2)$
 1: $\mathcal{I}_g \leftarrow \text{Grayscale}(I)$                                       ▷ Converts $I$ to grayscale
 2: $\mathcal{I}_b \leftarrow \text{Threshold}(I_g)$         ▷ Thresholds the grayscale image to produce a binary mask of the shape
 3: $\mathcal{C} \leftarrow \text{ExtractContours}(I_b)$                              ▷ Extract contour points from $I_b$
 4: $D \leftarrow \text{ApplyMATC}(\mathcal{C})$                        ▷ Apply Modified Adaptive Tangential Cover on $\mathcal{C}$
 5: **return** $D$                                                 ▷ Return the matrix of dominant points
---

## 3.2  Networks

**Baseline.** To compare with our approach, we used ResNet (He et al., 2016a) architecture as a baseline CNN which involves images as input. Although the ResNet use RGB images, this has been trained on the same data used to extract the dominant points for our approach. This ensures that the training is conducted on identical data and the metrics are calculated consistently. For the different experiments we evaluated several versions of ResNet (18, 34, 50) and have chosen to report on the version that delivered the best performance. While ViTs Dosovitskiy et al. (2020) have shown remarkable performance in various domains especially when it comes to large datasets, we selected ResNet as our baseline due to its well-established architecture, ease of implementation, and lower computational requirements. Additionally, ResNet-50 is more suitable for scenarios with limited data, ensuring a more straightforward and fair comparison for evaluating the effectiveness of our approach. For the different experiments ResNet uses 3 channel images and takes advantage of the rich color information in RGB images, which enhances feature representation by capturing detailed color variations, textures, and contextual cues that are often critical for distinguishing between visually similar objects.

**PolygoNet** An overview of the approach is depicted in Figure 2. To address the challenge of processing dominant points of varying lengths extracted from the original input images, our approach introduces a novel

adaptation of the self-attention mechanism, inspired by the transformer model from the pioneering work of Dosovitskiy et al. (2020). This methodology allows our model to focus dynamically across the input space and effectively process sets of points regardless of their quantity. By leveraging the self-attention capabilities, the model assigns appropriate weights to each dominant point, capturing the geometric nuances inherent to the dataset. In self-attention mechanisms, the model computes attention scores using the scaled dot-product of queries, keys, and values, which allows it to weigh the importance of each input token relative to others. This strategy ensures that the features extracted are reflective of the essential characteristics of the geometric figures, facilitating the consolidation of critical information within a structured feature space for subsequent analysis.

Our model further refines the extracted features through the incorporation of 1D convolutional blocks. These blocks serve to process the feature vector output from the attention mechanism, enhancing the model's ability to discern intricate geometric patterns within the dominant points data.

Table 1: PolygoNet architecture. Conv1d($n$) denotes a 1D convolutional layer with $n$ output channels. Each convolution is followed by batch normalization and a ReLU activation function. N denotes the number of dominant points

| Input tensor of shape (Features, Sequence Length) | |
|---|---|
| Input size | (N, 2) |
| Attention | Custom attention mechanism |
| $Layer_1$: | Conv1d(64), BatchNorm, ReLU, Dropout(0.1) |
| $Layer_2$: | Conv1d(128), BatchNorm, ReLU |
| $Layer_3$: | Conv1d(256), BatchNorm, ReLU |
| $Layer_4$: | Conv1d(512), BatchNorm, ReLU |
| $Layer_5$: | Conv1d(1024), BatchNorm, ReLU |
| Output: | Classifcation head with $num\_classes$, followed by activation |
| Output tensor of shape ($num\_classes$) | |

The architecture comprises layers of multiheaded self-attention (MSA), as utilized in Dosovitskiy et al. (2020), and Conv1D blocks. Normalization layer is applied before each block. Our model fundamentally enhances the traditional multi-head attention mechanism to better cater to the geometrical properties of data. For the different experiments, $f_\theta$ showed in Figure 2 represents a block of 2 Conv1D, each layer is followed by a Normalization layer and ReLU as activation function. The MLP Head represents a simple linear layer with number classes as a parameter.

The use of 1D convolutional (Conv1D) layers is particularly effective in this context due to their capacity for capturing local dependencies and patterns along the sequence of points and for computational efficiency, thereby augmenting the attention mechanism's global perspective with localized feature extraction. This sequential application of self-attention followed by Conv1D processing allows our model to enhance model's performance by effectively capturing both global dependencies and local patterns within the dominant point coordinates. The proposed method integrates global attention mechanisms with localized convolutional processing to effectively extract variable-length geometric features, addressing associated challenges with improved precision and robustness.

Positional embeddings are incorporated with dominant points coordinates to preserve positional data. In the context of our approach, the positional embedding refers to the ordered sequence that defines the form and structure of the shapes, enabling the model to incorporate the sequential arrangement into its understanding and processing. There are several choices of positional embedding, our method uses 1D learnable positional embedding as a standard approach which is based on the sine and cosine function of different frequencies Vaswani et al. (2017).

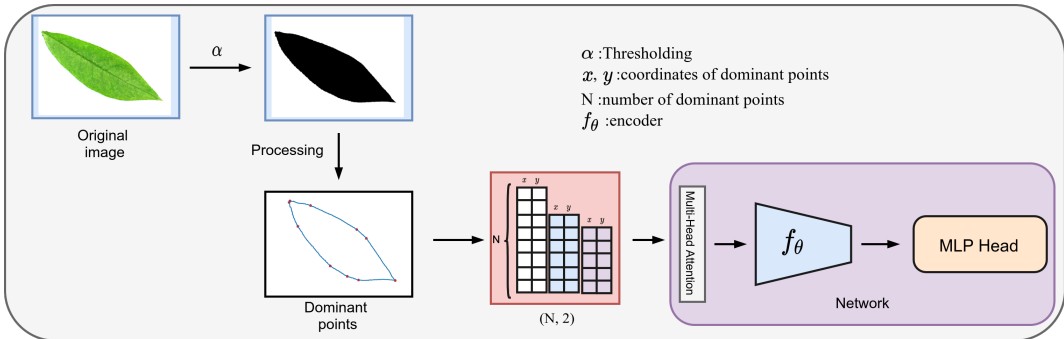

Figure 2: PolygoNet pipeline. The input colored image is converted to grayscale before being thresholded with Otsu. The dominant points are extracted using the MATC approach from the extracted contour. This variable size sequence of dominant points is then processed for classification by PolygoNet. Note that the complexity of the contour impacts the number of computed dominant points that will be processed by PolygoNet.

This integration is achieved using a standard approach based on *sine* and *cosine* functions, which provide unique positional encoding for different positions. This enables the model to distinguish between points based on their position in the sequence. Each position *pos* in the sequence is encoded using a combination of *sine* and *cosine* functions of varying frequencies, allowing the model to capture both the absolute and relative positions of the points. The positional encoding for a given position *pos* and dimension *i* is defined as follows:

$$PE_{(pos,2i)} = \sin\left(\frac{pos}{10000^{2i/d}}\right) \tag{1}$$

$$PE_{(pos,2i+1)} = \cos\left(\frac{pos}{10000^{2i/d}}\right) \tag{2}$$

By leveraging these positional encoding, our model can effectively retain the sequential and spatial relationships among the dominant points, enhancing its ability to capture the geometric the structure of the shapes.

## 4 Experiments

In this section, we explore the usage of our proposed approach for image classification task. We show results on three different datasets.

### 4.1 Setup

**Dataset.** To explore the model performances and robustness, we use several image classification datasets: FashionMNIST (Xiao et al., 2017) with 10 classes and 70.000 grayscale ($28 \times 28$) images, Flavia dataset (Wu et al., 2007) with 32 leaf classes and 1.9k colored images where each image has a resolution of ($1600 \times 1200$), Folio dataset (Munisami et al., 2015) with 32 plant classes, each class contains 20 RGB images with a resolution of ($4160 \times 3120$). For all these datasets, the objet to be classified is well segmented on a uniform background making it easy to extract the contour and further apply our pipeline. The Flavia dataset is particularly challenging due to the subtle differences between classes. The Folio dataset contains images of 32 different types of leaves taken from different plants with varying lighting conditions and scales. This dataset tests the adaptability and effectiveness of the model under less controlled imaging conditions.

**Implementation details** For all the experiments presented in the paper, we use Adam (Kingma & Ba, 2014) as optimizer with $\beta_1 = 0.9$ and $\beta_2 = 0.999$, a base learning rate of $10^{-5}$, we apply a weight decay of 0.0001. The loss is cross-entropy. For regularization, a dropout layer is inserted with a probability of

10% to mask a neuron. For the ResNet-50 architecture, which processes image data, we employed a series of data augmentation techniques including rotations, flipping. With our approach PolygoNet, which deals with dominant points coordinates, we used data augmentation applied for coordinates, such as rotation and flipping of coordinates. The baseline network is trained for 150 epochs, in contrast, our model is trained for 300 epochs. We implement the proposed approach with PyTorch Ansel et al. (2024), using a single NVIDIA RTX 3090, and we used CPU for some experiments to highlight the effectiveness of the proposed framework. For the different experiments we use early-stopping, and report the best validation metrics achieved during training. For the edge computing part, we used a Nvidia Jetson Orin Nano.

**Metrics.** For the different experiments, we used two mainly quantitative metrics to evaluate the quality and the performance of our approach including accuracy and F1-score. By including the F1-score in our evaluation metrics, we ensure that our model's performance was thoroughly assessed, taking into account the balance between precision and recall.

## 5 Results

In this section, we evaluate our approach using three different datasets and compare it to a baseline model. The datasets used for this evaluation include FashionMNIST, Flavia, and Folio. Each dataset tests the robustness and efficiency of our model across diverse domains. The results on the test fold are summarized in Table 2. Additionally, we compare the processing time including the inference time in different configuration such as server and edge computing systems.

Table 2: Comparison of Model Performance on Various Datasets

| Dataset | Method | F1-score ↑ | Accuracy ↑ | FLOPs ↓ |
|---|---|---|---|---|
| FashionMNIST | Our | 0.90 | 0.78 | **8.52 M** |
| | ResNet-50 | 0.93 | 0.90 | 80.38 M |
| Flavia | Our | 0.90 | 0.79 | **8.67 M** |
| | ResNet-50 | 0.90 | 0.91 | 21.47 G |
| Folio | Our | 0.88 | 0.78 | **8.66 M** |
| | ResNet-50 | 0.84 | 0.86 | 21.47 G |

### 5.1 Evaluation on FashionMNIST Dataset

In our experiments, we used a standard split of $60,000$ training images and $10,000$ test images. The baseline model was trained with a batch size of 64 for 150 epochs. Our model was trained with the same batch size for 300 epochs. PolygoNet, achieved a F1-score of 0.90 and an accuracy of 78%, which are particularly noteworthy given the model's lower computational complexity, which is quantified at 8.52 million FLOPs. In contrast, ResNet-50, a more computationally intensive model, achieved a higher accuracy of 90% but at a significantly greater computational cost of 80.38 million FLOPs. The F1-score value for ResNet-50 on FashionMNIST was 0.93, indicating a slightly better ability to match the class labels accurately.

### 5.2 Evaluation on Flavia Dataset

For the baseline model we resized the images to 512x512 to fit in GPU. PolygoNet demonstrated robust performance on the Flavia dataset with a F1-score of 0.90 and an accuracy of 79%, while keeping the computational cost relatively low at 8.67 million FLOPs. The ResNet-50 model, although achieving a higher accuracy of 91% and a F1-score of 0.90, again showed its computationally expensive nature with 21.47 billion FLOPs

### 5.3 Evaluation on Folio Dataset

On the Folio dataset, PolygoNet achieved a F1-score of 0.88 and an accuracy of 78% with only 8.66 million FLOPs, showcasing its efficiency. The more resource-intensive ResNet-50 managed a slightly higher accuracy of 86% and a F1-score of 0.84, but at a cost of 21.47 billion FLOPs.

### 5.4 Evaluation of Processing time

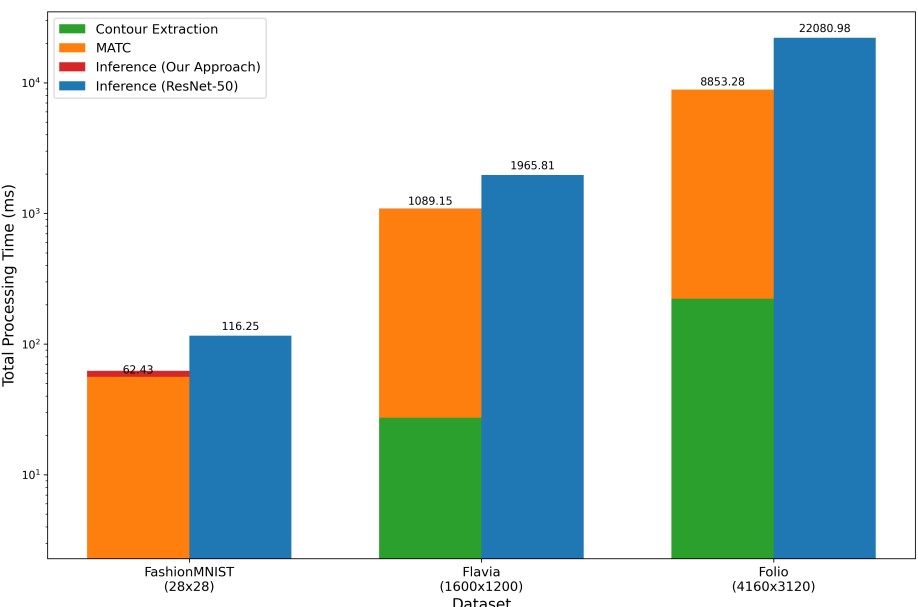

Figure 3: Inference Time by Dataset and Approach on Edge computing configuration. The plot compares the inference times for three different datasets using two approaches: Our Approach (orange) and ResNet-50 (blue). The y-axis is on a logarithmic scale to highlight the differences in performance.

The benchmarking results for processing time of the two pipelines, PolygoNet and ResNet-50, across three datasets (FashionMNIST, Folio, and Flavia) and two device configurations (Server and Edge Computing) are summarized in Table 3. These results provide a comprehensive comparison of the computational efficiency and practicality of each pipeline under different configurations.

For the FashionMNIST dataset, PolygoNet significantly outperforms ResNet-50 in terms of total processing time. On the server configuration, PolygoNet completes the entire process in 9.66 ms, whereas ResNet-50 requires 17.06 ms. This trend is even more pronounced in edge computing, where PolygoNet takes 62.43 ms, while ResNet-50 demands a substantial 116.25 ms. The efficient contour extraction and MATC processing steps of PolygoNet contribute to its lower overall processing time.

For the Folio dataset, PolygoNet showcases a remarkable reduction in processing time, highlighting its superior efficiency compared to ResNet-50. This difference is even more stark on edge computing devices, underscoring PolygoNet's suitability for resource-constrained environments.

Similarly, for the Flavia dataset, PolygoNet demonstrates superior efficiency over ResNet-50 on both server and edge computing devices. This consistent performance advantage across datasets and device configurations highlights PolygoNet's ability to deliver rapid and efficient processing.

These results emphasize the practical benefits of PolygoNet, especially in scenarios where computational resources and processing time are critical constraints. The significant reduction in total processing time, particularly on edge computing devices, makes PolygoNet an attractive solution for real-time applications in mobile and embedded systems. The efficient handling of dominant points and the streamlined processing

pipeline enable PolygoNet to maintain high performance while conserving computational resources, showcasing its potential for widespread adoption in various real-world AI applications.

Table 3: Benchmarking Processing Time of Two Pipelines on Three Datasets Across Two Configuration

| Dataset | Device | Pipeline | Contour Extract (ms) | MATC (ms) | Inference (ms) | Total Time (ms) |
|---|---|---|---|---|---|---|
| FashionMNIST ($28 \times 28$) | Workstation | Our | 1.68 | 6.22 | 1.76 | **9.66** |
| | | ResNet-50 | - | - | 17.06 | 17.06 |
| | Edge Computing | Our | 2.28 | 54 | 6.15 | **62.43** |
| | | ResNet-50 | - | - | 116.25 | 116.25 |
| Flavia ($1600 \times 1200$) | Workstation | Our | 13.80 | 125 | 1.51 | **140.31** |
| | | ResNet-50 | - | - | 276.87 | 276.87 |
| | Edge Computing | Our | 27.38 | 1054 | 7.77 | **1089.15** |
| | | ResNet-50 | - | - | 1965.81 | 1965.81 |
| Folio ($4160 \times 3120$) | Workstation | Our | 104.27 | 848 | 4.30 | **956.57** |
| | | ResNet-50 | - | - | 2073.29 | 2073.29 |
| | Edge Computing | Our | 223 | 8622 | 8.28 | **8853.28** |
| | | ResNet-50 | - | - | 22080.98 | 22080.98 |

## 5.5 Discussions

Across all datasets, PolygoNet consistently required fewer floating-point operations (FLOPs), highlighting its suitability for applications where computational resources are limited. Although ResNet-50 often achieved higher accuracy and F1-score, the trade-off in terms of computational demand makes PolygoNet a more practical choice in resource-constrained environments. This evaluation demonstrates the capability of PolygoNet to provide a balance between performance and computational efficiency, which is critical for real-world applications, particularly in mobile and embedded systems.

In evaluating PolygoNet's performance across various datasets, it is evident that the model's design strategically balances computational efficiency with sufficient accuracy for practical applications. Particularly in environments where computational resources are scarce, such as in mobile and embedded systems, PolygoNet's low FLOP count not only conserves energy but also enables more widespread use of advanced AI technologies. This balance is crucial for applications requiring real-time analytics, such as in autonomous vehicles or remote sensing technologies. Future research could focus on enhancing PolygoNet's efficiency further through techniques such as quantization and pruning, which may reduce the model size without significantly impacting accuracy. Additionally, extending PolygoNet's capabilities to handle more complex tasks or datasets without substantial increases in computational demands could significantly impact AI's accessibility and sustainability.

## 6 Conclusion

In this paper, we have presented PolygoNet, a new approach that leverages polygonal contours and dominant points for efficient image classification using deep neural networks. Our methodology addresses key challenges in image classification, such as computational complexity and hardware resources requirement, by transforming input images into compact polygon representations. This transformation significantly reduces the computational burden, making our approach suitable for real-time applications and resource-constrained environments.

The experimental results on benchmark datasets, including FashionMNIST, Flavia, and Folio, demonstrate that PolygoNet achieves state-of-the-art performance while maintaining low computational requirements. PolygoNet's design, which combines the Modified Adaptive Tangential Cover (MATC) for dominant point extraction with self-attention mechanisms and 1D convolutional blocks, ensures robust and efficient processing of image data. This combination effectively balances model performance with the capacity to generalize from minimal data while adhering to computational constraints.

Our evaluations highlight that PolygoNet offers a balanced trade-off between accuracy and computational efficiency. While traditional models like ResNet-50 achieve higher accuracy, they come with significantly

higher computational costs. In contrast, PolygoNet provides comparable performance with a fraction of the computational resources, making it a practical choice for edge computing and mobile AI applications.

Future research could explore further optimization techniques, such as model quantization and pruning, to enhance PolygoNet's efficiency. Additionally, extending the approach to handle more complex datasets and tasks without increasing computational demands could broaden its applicability. The promising results of PolygoNet pave the way for its integration into various real-world applications, where efficient and accurate image classification is crucial.

Overall, PolygoNet represents a good trade-off between performance and computational efficiency in image classification, demonstrating that it is possible to achieve high performance with reduced computational overhead by focusing on essential geometric features. This work opens new avenues for developing resource-efficient deep learning models capable of performing complex visual tasks in constrained environments.

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
