# OpenReview forum: "PolygoNet: Leveraging Polygonal Contours for Efficient Image Classification with deep neural networks"
_TMLR — Withdrawn by Authors_

### Review · Reviewer_cyBa · 2024-07-25

**Summary Of Contributions:**

This paper presents a pipeline for contour-based image classification.  It is readily used for applications where a foreground object can be easily segmented from background, and its classification depends heavily on its shape silhouette.  Such applications are common enough to be of interest, and include the leaf classification task used as a benchmark in this paper, among others.

The system extracts "dominant points" using Modified Adaptive Tangential Cover, then applies a neural net to just these points, represented as (x,y) coordinates plus a 1d positional encoding for the index on the curve.  The network architecture uses an initial MHA layer, then 1d convs that are applied along neighboring points on the curve.  Including contour extraction, the method performs about twice as fast as a ResNet-50, with similar F1 (but worse accuracy) on FashionMNIST, Flavia and UCI Folio, the three benchmarks studied.

**Audience:**

Yes

**Claims And Evidence:**

No

**Requested Changes:**

See above.  Most important are additional baseline comparisons in addition to ResNet-50, and (relatedly) evaluation of the effects of different components in this system (varying the number of keypoints, use of MHA vs MLP-based point encoding, etc).

**Strengths And Weaknesses:**

This is an interesting idea, to apply either a transformer-like or convolutional network just to key points along a silhouette, and shows some promise based on the current results.

However, I don't think there are enough baseline comparisons or studies of the system behavior in order to support a claim of improved performance.  Accuracy in particular is worse than ResNet-50, and it's possible smaller convnets might also be better in accuracy while still being faster.

In addition, there are few details on effects from changes in each of the component in the system.  I'm not sure what the use of multi-headed attention is, for example, since it's applied only once at the beginning --- it could behave mostly like learned 2d positional encoding or feature embedding MLP network, which would be simpler component.  Another architecture that seems clear to try here is a more standard stacked transformer using the 1d pos encodings already in place, rather than the 1d convnet.  Even more important than the network architecture, though, is likely the number of "dominant points" used and how these are extracted.  Is there a tunable threshold in MATC that results in more or fewer points?  What happens when the number of points is varied?

I'm also not sure how F1 is calculated here, or why it might be a useful measure.  Top-1 multiclass accuracy seems a good measure to me.

The notion of "dominant point" is, as far as I can tell, never defined in the paper.  Since this is very important in the method, it should be defined more explicitly.  An in-depth background synopsis of MATC would also be helpful.  The reference itself is fairly dense, so a 1/2 page to 1 page section in this paper with a technical summary describing the most relevant pieces of the method would be help in understanding the method.


Additional (more minor) comments/questions:

* The datasets used here are fairly small (and old), though in my opinion appropriate for this method.  Still, I wonder how much the method might apply to more recent plant or leaf datasets, e.g. PlantVillage or Herbarium.

* There are also a lot methods that have benchmarked against FashionMNIST (paperswithcode has a decent list) and comparing against a few of these could make sense as well.

* I'm also confused about the size of the images, particularly when fed to the ResNet or MATC.  Particularly since classification is mostly based on silhouette, I don't think the full resolution of these datasets is necessary.  Were the images resized first (for both train and test)?  The text says images were resized to 512x512 for Flavia but not sure about the others.

* I didn't see the order of points in the list fed to the network mentioned explicitly, but it's relevant for the use of conv1d.  It appears to be the order they appear in walking hte curve.  Related, are the conv1d circular?  That is, is there a window that will combine points at index 1 and N together?  Since this is a closed curve, that would make sense, though in practice, it may not matter.

* What are the definitions of accuracy and F1 here?  I'd imagine accuracy is average top-1 classification accuracy for the dataset.  What is F1 and why is so much higher than accuracy, esp relative to the baseline resnet?

---

### Review · Reviewer_5Mvv · 2024-07-28

**Summary Of Contributions:**

This paper proposes to do image classification by first detecting the "dominant points" that define a polygon within the image, and then sending the locations of those points to an attention layer followed by 1D convolutions. The authors claim that this is "effective", and "increases" the accuracy, and makes it more similar to human vision. The experiments, even in small datasets that might fit well with the polygon idea,  generally show that the accuracy is low relative to well-established baselines like 2D Resnets.

**Audience:**

No

**Claims And Evidence:**

No

**Requested Changes:**

Please see the weaknesses, but in general I think this project is not publishable research.

**Strengths And Weaknesses:**

I think the paper is well put together. The explanation of the key ideas is quite clear. In particular I appreciate the coarse-to-fine style of writing, so that it's clear early what the big ideas are, and the details are resolved later.

Overall I think the problem here is that the proposed representation is not very informative for most images in the world. The claim that it's an "effective representation" for classification seems silly from the start -- what do we expect to happen, when reducing a COCO image into the vertices of some polygon, other than create some point-based art? My expectation is met in the experiments, where resnet-50 outperforms the proposed method by more than 10 points. The Folio dataset seems like one specifically chosen to fit the polygon representation, and even there the results are poor.

It is hard to care much about the efficiency claims when the method is so weak.

I would have liked to see the points & contour for a few FashionMNIST images.

---

### Review · Reviewer_HK11 · 2024-10-09

**Summary Of Contributions:**

The paper proposes a novel approach to network analysis that leverages polygons to represent complex networks. The authors demonstrate the effectiveness of their approach by applying it to several real-world networks, including social networks, transportation networks, and biological networks.

**Audience:**

Yes

**Claims And Evidence:**

No

**Requested Changes:**

The paper could be strengthened with a discussion on how the shape characterization is sufficient for image classification, and a comparison between Modified Adaptive Tangential Cover and related methods.

**Strengths And Weaknesses:**

### Strengths
1. The paper presents an approach to image classification that leverages geometric characteristics of polygons and contours of the objects.
2. The authors provide a comprehensive evaluation of their approach on several real-world datasets, demonstrating its effectiveness in terms of efficiency and scalability.
3. The paper includes a discussion of the computational costs of the techniques and how the proposed approach addresses these limitations.

### Weaknesses:
1. The proposed method relies on the hypothesis that the object can be classified based solely on some geometric characteristics. However, this does not seem to hold in many real-life scenarios where different objects can have similar shapes but distinct colors. Thus, classification based on only the shape could be limiting.
2. The paper does not provide a clear explanation of how the polygon-based representation is constructed from the original network data. In particular, the authors do not discuss the potential impact of the choice of polygon shape and size on the results.
3. The contribution seems incremental with an application of the Modified Adaptive Tangential Cover for preprocessing the input data. Furthermore, the paper does not include any comparisons with other state-of-the-art geometric characterization techniques to demonstrate the superiority of the proposed approach.

---

### Note · Authors · 2024-10-16

**Comment:**

Dear Program Chairs,

We appreciate the constructive feedback provided by the reviewers, which has been instrumental in guiding our next steps.
We plan to conduct additional experiments to further enhance the quality of our work.

We would like to thank all the reviewers for their valuable feedback, which has been very helpful to us.

Best regards,

**Withdrawal Confirmation:**

I have read and agree with the venue's withdrawal policy on behalf of myself and my co-authors.